# Top-k Multiclass SVM

**Maksim Lapin,**[1] **Matthias Hein**[2] **and Bernt Schiele**[1]
[1]Max Planck Institute for Informatics, Saarbrücken, Germany
[2]Saarland University, Saarbrücken, Germany

## Abstract

Class ambiguity is typical in image classification problems with a large number of classes. When classes are difficult to discriminate, it makes sense to allow $k$ guesses and evaluate classifiers based on the top-$k$ error instead of the standard zero-one loss. We propose top-$k$ multiclass SVM as a direct method to optimize for top-$k$ performance. Our generalization of the well-known multiclass SVM is based on a tight convex upper bound of the top-$k$ error. We propose a fast optimization scheme based on an efficient projection onto the top-$k$ simplex, which is of its own interest. Experiments on five datasets show consistent improvements in top-$k$ accuracy compared to various baselines.

## 1   Introduction

As the number of classes increases, two important issues emerge: class overlap and multi-label nature of examples [9]. This phenomenon asks for adjustments of both the evaluation metrics as well as the loss functions employed. When a predictor is allowed $k$ guesses and is not penalized for $k-1$ mistakes, such an evaluation measure is known as top-$k$ error. We argue that this is an important metric that will inevitably receive more attention in the future as the illustration in Figure 1 indicates.

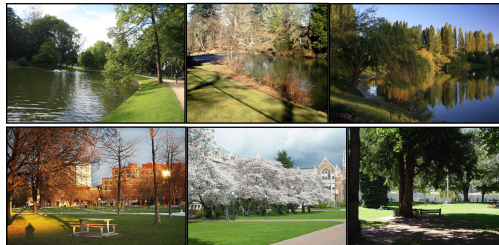

Figure 1: Images from SUN 397 [29] illustrating class ambiguity. **Top:** (left to right) Park, River, Pond. **Bottom:** Park, Campus, Picnic area.

How obvious is it that each row of Figure 1 shows examples of *different* classes? Can we imagine a human to predict correctly on the first attempt? Does it even make sense to penalize a learning system for such "mistakes"? While the problem of class ambiguity is apparent in computer vision, similar problems arise in other domains when the number of classes becomes large.

We propose top-$k$ multiclass SVM as a generalization of the well-known multiclass SVM [5]. It is based on a tight convex upper bound of the top-$k$ zero-one loss which we call **top-$k$ hinge loss**. While it turns out to be similar to a top-$k$ version of the ranking based loss proposed by [27], we show that the top-$k$ hinge loss is a lower bound on their version and is thus a tighter bound on the top-$k$ zero-one loss. We propose an efficient implementation based on stochastic dual coordinate ascent (SDCA) [24]. A key ingredient in the optimization is the (biased) projection onto the top-$k$ simplex. This projection turns out to be a tricky generalization of the continuous quadratic knapsack problem, respectively the projection onto the standard simplex. The proposed algorithm for solving it has complexity $O(m \log m)$ for $x \in \mathbb{R}^m$. Our implementation of the top-$k$ multiclass SVM scales to large datasets like Places 205 with about $2.5$ million examples and $205$ classes [30]. Finally, extensive experiments on several challenging computer vision problems show that top-$k$ multiclass SVM consistently improves in top-$k$ error over the multiclass SVM (equivalent to our top-1 multiclass SVM), one-vs-all SVM and other methods based on different ranking losses [11, 16].

## 2 Top-$k$ Loss in Multiclass Classification

In multiclass classification, one is given a set $S = \{(x_i, y_i) \,|\, i = 1, \ldots, n\}$ of $n$ training examples $x_i \in \mathcal{X}$ along with the corresponding labels $y_i \in \mathcal{Y}$. Let $\mathcal{X} = \mathbb{R}^d$ be the feature space and $\mathcal{Y} = \{1, \ldots, m\}$ the set of labels. The task is to learn a set of $m$ linear predictors $w_y \in \mathbb{R}^d$ such that the risk of the classifier $\arg\max_{y \in \mathcal{Y}} \langle w_y, x \rangle$ is minimized for a given loss function, which is usually chosen to be a convex upper bound of the zero-one loss. The generalization to nonlinear predictors using kernels is discussed below.

The classification problem becomes extremely challenging in the presence of a large number of ambiguous classes. It is natural in that case to extend the evaluation protocol to allow $k$ guesses, which leads to the popular top-$k$ error and top-$k$ accuracy performance measures. Formally, we consider a ranking of labels induced by the prediction scores $\langle w_y, x \rangle$. Let the bracket $[\cdot]$ denote a permutation of labels such that $[j]$ is the index of the $j$-th largest score, i.e.

$$\langle w_{[1]}, x \rangle \geq \langle w_{[2]}, x \rangle \geq \ldots \geq \langle w_{[m]}, x \rangle.$$

The top-$k$ zero-one loss $\mathrm{err}_k$ is defined as

$$\mathrm{err}_k(f(x), y) = \mathbb{1}_{\langle w_{[k]}, x \rangle > \langle w_y, x \rangle},$$

where $f(x) = (\langle w_1, x \rangle, \ldots, \langle w_m, x \rangle)^\top$ and $\mathbb{1}_P = 1$ if $P$ is true and $0$ otherwise. Note that the standard zero-one loss is recovered when $k = 1$, and $\mathrm{err}_k(f(x), y)$ is always $0$ for $k = m$. Therefore, we are interested in the regime $1 \leq k < m$.

### 2.1 Multiclass Support Vector Machine

In this section we review the multiclass SVM of Crammer and Singer [5] which will be extended to the top-$k$ multiclass SVM in the following. We mainly follow the notation of [24].

Given a training pair $(x_i, y_i)$, the multiclass SVM loss on example $x_i$ is defined as

$$\max_{y \in \mathcal{Y}} \{ \mathbb{1}_{y \neq y_i} + \langle w_y, x_i \rangle - \langle w_{y_i}, x_i \rangle \}. \tag{1}$$

Since our optimization scheme is based on Fenchel duality, we also require a convex conjugate of the primal loss function (1). Let $c \triangleq \mathbf{1} - e_{y_i}$, where $\mathbf{1}$ is the all ones vector and $e_j$ is the $j$-th standard basis vector in $\mathbb{R}^m$, let $a \in \mathbb{R}^m$ be defined componentwise as $a_j \triangleq \langle w_j, x_i \rangle - \langle w_{y_i}, x_i \rangle$, and let

$$\Delta \triangleq \{ x \in \mathbb{R}^m \,|\, \langle \mathbf{1}, x \rangle \leq 1, \ 0 \leq x_i, \ i = 1, \ldots, m \}.$$

**Proposition 1** ([24], § 5.1). *A primal-conjugate pair for the multiclass SVM loss (1) is*

$$\phi(a) = \max\{0, (a + c)_{[1]}\}, \qquad \phi^*(b) = \begin{cases} -\langle c, b \rangle & \text{if } b \in \Delta, \\ +\infty & \text{otherwise.} \end{cases} \tag{2}$$

Note that thresholding with $0$ in $\phi(a)$ is actually redundant as $(a + c)_{[1]} \geq (a + c)_{y_i} = 0$ and is only given to enhance similarity to the top-$k$ version defined later.

### 2.2 Top-$k$ Support Vector Machine

The main motivation for the top-$k$ loss is to relax the penalty for making an error in the top-$k$ predictions. Looking at $\phi$ in (2), a direct extension to the top-$k$ setting would be a function

$$\psi_k(a) = \max\{0, (a + c)_{[k]}\},$$

which incurs a loss iff $(a + c)_{[k]} > 0$. Since the ground truth score $(a + c)_{[y_i]} = 0$, we conclude that

$$\psi_k(a) > 0 \iff \langle w_{[1]}, x_i \rangle \geq \ldots \geq \langle w_{[k]}, x_i \rangle > \langle w_{y_i}, x_i \rangle - 1,$$

which directly corresponds to the top-$k$ zero-one loss $\mathrm{err}_k$ with margin $1$.

Note that the function $\psi_k$ ignores the values of the first $(k - 1)$ scores, which could be quite large if there are highly similar classes. That would be fine in this model as long as the correct prediction is

within the first $k$ guesses. However, the function $\psi_k$ is unfortunately nonconvex since the function $f_k(x) = x_{[k]}$ returning the $k$-th largest coordinate is nonconvex for $k \geq 2$. Therefore, finding a globally optimal solution is computationally intractable.

Instead, we propose the following convex upper bound on $\psi_k$, which we call the **top-$k$ hinge loss**,

$$\phi_k(a) = \max\left\{0, \frac{1}{k}\sum_{j=1}^{k}(a+c)_{[j]}\right\}, \tag{3}$$

where the sum of the $k$ largest components is known to be convex [3]. We have that

$$\psi_k(a) \leq \phi_k(a) \leq \phi_1(a) = \phi(a),$$

for any $k \geq 1$ and $a \in \mathbb{R}^m$. Moreover, $\phi_k(a) < \phi(a)$ unless all $k$ largest scores are the same. This extra slack can be used to increase the margin between the current and the $(m - k)$ remaining least similar classes, which should then lead to an improvement in the top-$k$ metric.

### 2.2.1 Top-$k$ Simplex and Convex Conjugate of the Top-$k$ Hinge Loss

In this section we derive the conjugate of the proposed loss (3). We begin with a well known result that is used later in the proof. All proofs can be found in the supplement. Let $[a]_+ = \max\{0, a\}$.

**Lemma 1** ([17], Lemma 1). $\sum_{j=1}^{k} h_{[j]} = \min_t \left\{kt + \sum_{j=1}^{m}[h_j - t]_+\right\}$.

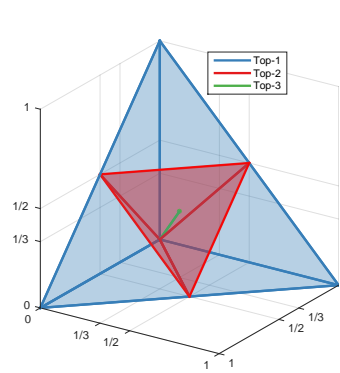

Figure 2: Top-$k$ simplex $\Delta_k(1)$ for $m = 3$. Unlike the standard simplex, it has $\binom{m}{k} + 1$ vertices.

We also define a set $\Delta_k$ which arises naturally as the effective domain[1] of the conjugate of (3). By analogy, we call it the top-$k$ simplex as for $k = 1$ it reduces to the standard simplex with the inequality constraint (i.e. $0 \in \Delta_k$). Let $[m] \triangleq 1, \ldots, m$.

**Definition 1.** *The* top-$k$ simplex *is a convex polytope defined as*

$$\Delta_k(r) \triangleq \left\{x \,\middle|\, \langle \mathbf{1}, x\rangle \leq r, \; 0 \leq x_i \leq \frac{1}{k}\langle \mathbf{1}, x\rangle, \; i \in [m]\right\},$$

*where $r \geq 0$ is the bound on the sum $\langle \mathbf{1}, x\rangle$. We let $\Delta_k \triangleq \Delta_k(1)$.*

The crucial difference to the standard simplex is the upper bound on $x_i$'s, which limits their maximal contribution to the total sum $\langle \mathbf{1}, x\rangle$. See Figure 2 for an illustration.

The first technical contribution of this work is as follows.

**Proposition 2.** *A primal-conjugate pair for the top-$k$ hinge loss (3) is given as follows:*

$$\phi_k(a) = \max\left\{0, \frac{1}{k}\sum_{j=1}^{k}(a+c)_{[j]}\right\}, \qquad \phi_k^*(b) = \begin{cases} -\langle c, b\rangle & \text{if } b \in \Delta_k, \\ +\infty & \text{otherwise.} \end{cases} \tag{4}$$

*Moreover, $\phi_k(a) = \max\{\langle a+c, \lambda\rangle \mid \lambda \in \Delta_k\}$.*

Therefore, we see that the proposed formulation (3) naturally extends the multiclass SVM of Crammer and Singer [5], which is recovered when $k = 1$. We have also obtained an interesting extension (or rather contraction, since $\Delta_k \subset \Delta$) of the standard simplex.

### 2.3 Relation of the Top-$k$ Hinge Loss to Ranking Based Losses

Usunier et al. [27] have recently formulated a very general family of convex losses for ranking and multiclass classification. In their framework, the hinge loss on example $x_i$ can be written as

$$L_\beta(a) = \sum_{y=1}^{m} \beta_y \max\{0, (a+c)_{[y]}\},$$

where $\beta_1 \geq \ldots \geq \beta_m \geq 0$ is a non-increasing sequence of non-negative numbers which act as weights for the ordered losses.

The relation to the top-$k$ hinge loss becomes apparent if we choose $\beta_j = \frac{1}{k}$ if $j \leq k$, and 0 otherwise. In that case, we obtain another version of the top-$k$ hinge loss

$$\tilde{\phi}_k(a) = \frac{1}{k} \sum_{j=1}^{k} \max\{0, (a+c)_{[j]}\}. \tag{5}$$

It is straightforward to check that

$$\psi_k(a) \leq \phi_k(a) \leq \tilde{\phi}_k(a) \leq \phi_1(a) = \tilde{\phi}_1(a) = \phi(a).$$

The bound $\phi_k(a) \leq \tilde{\phi}_k(a)$ holds with equality if $(a+c)_{[1]} \leq 0$ or $(a+c)_{[k]} \geq 0$. Otherwise, there is a gap and our top-$k$ loss is a strictly better upper bound on the actual top-$k$ zero-one loss. We perform extensive evaluation and comparison of both versions of the top-$k$ hinge loss in § 5.

While [27] employed LaRank [1] and [9], [28] optimized an approximation of $L_\beta(a)$, we show in the supplement how the loss function (5) can be optimized exactly and efficiently within the Prox-SDCA framework.

**Multiclass to binary reduction.** It is also possible to compare directly to ranking based methods that solve a binary problem using the following reduction. We employ it in our experiments to evaluate the ranking based methods SVM$^{\mathrm{Perf}}$ [11] and TopPush [16]. The trick is to augment the training set by embedding each $x_i \in \mathbb{R}^d$ into $\mathbb{R}^{md}$ using a feature map $\Phi_y$ for each $y \in \mathcal{Y}$. The mapping $\Phi_y$ places $x_i$ at the $y$-th position in $\mathbb{R}^{md}$ and puts zeros everywhere else. The example $\Phi_{y_i}(x_i)$ is labeled $+1$ and all $\Phi_y(x_i)$ for $y \neq y_i$ are labeled $-1$. Therefore, we have a new training set with $mn$ examples and $md$ dimensional (sparse) features. Moreover, $\langle w, \Phi_y(x_i) \rangle = \langle w_y, x_i \rangle$ which establishes the relation to the original multiclass problem.

Another approach to general performance measures is given in [11]. It turns out that using the above reduction, one can show that under certain constraints on the classifier, the recall@$k$ is equivalent to the top-$k$ error. A convex upper bound on recall@$k$ is then optimized in [11] via structured SVM. As their convex upper bound on the recall@$k$ is not decomposable in an instance based loss, it is not directly comparable to our loss. While being theoretically very elegant, the approach of [11] does not scale to very large datasets.

## 3 Optimization Framework

We begin with a general $\ell_2$-regularized multiclass classification problem, where for notational convenience we keep the loss function unspecified. The multiclass SVM or the top-$k$ multiclass SVM are obtained by plugging in the corresponding loss function from § 2.

### 3.1 Fenchel Duality for $\ell_2$-Regularized Multiclass Classification Problems

Let $X \in \mathbb{R}^{d \times n}$ be the matrix of training examples $x_i \in \mathbb{R}^d$, let $W \in \mathbb{R}^{d \times m}$ be the matrix of primal variables obtained by stacking the vectors $w_y \in \mathbb{R}^d$, and $A \in \mathbb{R}^{m \times n}$ the matrix of dual variables.

Before we prove our main result of this section (Theorem 1), we first impose a technical constraint on a loss function to be compatible with the choice of the ground truth coordinate. The top-$k$ hinge loss from Section 2 satisfies this requirement as we show in Proposition 3. We also prove an auxiliary Lemma 2, which is then used in Theorem 1.

**Definition 2.** *A convex function $\phi$ is $j$-compatible if for any $y \in \mathbb{R}^m$ with $y_j = 0$ we have that*

$$\sup\{\langle y, x \rangle - \phi(x) \,|\, x_j = 0\} = \phi^*(y).$$

This constraint is needed to prove equality in the following Lemma.

**Lemma 2.** *Let $\phi$ be $j$-compatible, let $H_j = \mathbf{I} - \mathbf{1}e_j^\top$, and let $\Phi(x) = \phi(H_j x)$, then*

$$\Phi^*(y) = \begin{cases} \phi^*(y - y_j e_j) & \text{if } \langle \mathbf{1}, y \rangle = 0, \\ +\infty & \text{otherwise.} \end{cases}$$

We can now use Lemma 2 to compute convex conjugates of the loss functions.

**Theorem 1.** *Let $\phi_i$ be $y_i$-compatible for each $i \in [n]$, let $\lambda > 0$ be a regularization parameter, and let $K = X^\top X$ be the Gram matrix. The primal and Fenchel dual objective functions are given as:*

$$P(W) = +\frac{1}{n} \sum_{i=1}^{n} \phi_i \left( W^\top x_i - \langle w_{y_i}, x_i \rangle \mathbf{1} \right) + \frac{\lambda}{2} \operatorname{tr} \left( W^\top W \right),$$

$$D(A) = -\frac{1}{n} \sum_{i=1}^{n} \phi_i^* \left( -\lambda n (a_i - a_{y_i,i} e_{y_i}) \right) - \frac{\lambda}{2} \operatorname{tr} \left( AKA^\top \right), \ \text{if } \langle \mathbf{1}, a_i \rangle = 0 \ \forall i, \ +\infty \ \text{otherwise}.$$

*Moreover, we have that $W = XA^\top$ and $W^\top x_i = AK_i$, where $K_i$ is the $i$-th column of $K$.*

Finally, we show that Theorem 1 applies to the loss functions that we consider.

**Proposition 3.** *The top-$k$ hinge loss function from Section 2 is $y_i$-compatible.*

We have repeated the derivation from Section 5.7 in [24] as there is a typo in the optimization problem (20) leading to the conclusion that $a_{y_i,i}$ must be 0 at the optimum. Lemma 2 fixes this by making the requirement $a_{y_i,i} = -\sum_{j \neq y_i} a_{j,i}$ explicit. Note that this modification is already mentioned in their pseudo-code for Prox-SDCA.

### 3.2 Optimization of Top-$k$ Multiclass SVM via Prox-SDCA

As an optimization scheme, we employ the proximal stochastic dual coordinate ascent (Prox-SDCA) framework of Shalev-Shwartz and Zhang [24], which has strong convergence guarantees and is easy to adapt to our problem. In particular, we iteratively update a batch $a_i \in \mathbb{R}^m$ of dual variables corresponding to the training pair $(x_i, y_i)$, so as to maximize the dual objective $D(A)$ from Theorem 1. We also maintain the primal variables $W = XA^\top$ and stop when the relative duality gap is below $\epsilon$. This procedure is summarized in Algorithm 1.

Let us make a few comments on the advantages of the proposed method. First, apart from the update step which we discuss below, all main operations can be computed using a BLAS li-

---

**Algorithm 1** Top-$k$ Multiclass SVM

1: **Input:** training data $\{(x_i, y_i)_{i=1}^n\}$, parameters $k$ (loss), $\lambda$ (regularization), $\epsilon$ (stopping cond.)
2: **Output:** $W \in \mathbb{R}^{d \times m}$, $A \in \mathbb{R}^{m \times n}$
3: **Initialize:** $W \leftarrow 0$, $A \leftarrow 0$
4: **repeat**
5:     randomly permute training data
6:     **for** $i = 1$ **to** $n$ **do**
7:         $s_i \leftarrow W^\top x_i$ {prediction scores}
8:         $a_i^{\text{old}} \leftarrow a_i$ {cache previous values}
9:         $a_i \leftarrow update(k, \lambda, \|x_i\|^2, y_i, s_i, a_i)$
            {see § 3.2.1 for details}
10:         $W \leftarrow W + x_i (a_i - a_i^{\text{old}})^\top$
            {rank-1 update}
11:     **end for**
12: **until** relative duality gap is below $\epsilon$

---

brary, which makes the overall implementation efficient. Second, the update step in Line 9 is optimal in the sense that it yields maximal dual objective increase jointly over $m$ variables. This is opposed to SGD updates with data-independent step sizes, as well as to maximal but *scalar* updates in other SDCA variants. Finally, we have a well-defined stopping criterion as we can compute the duality gap (see discussion in [2]). The latter is especially attractive if there is a time budget for learning. The algorithm can also be easily kernelized since $W^\top x_i = AK_i$ (cf. Theorem 1).

#### 3.2.1 Dual Variables Update

For the proposed top-$k$ hinge loss from Section 2, optimization of the dual objective $D(A)$ over $a_i \in \mathbb{R}^m$ given other variables fixed is an instance of a regularized (biased) projection problem onto the top-$k$ simplex $\Delta_k(\frac{1}{\lambda n})$. Let $a^{\setminus j}$ be obtained by removing the $j$-th coordinate from vector $a$.

**Proposition 4.** *The following two problems are equivalent with $a_i^{\setminus y_i} = -x$ and $a_{y_i,i} = \langle \mathbf{1}, x \rangle$*

$$\max_{a_i}\{D(A) \mid \langle \mathbf{1}, a_i \rangle = 0\} \equiv \min_{x}\{\|b - x\|^2 + \rho \langle \mathbf{1}, x \rangle^2 \mid x \in \Delta_k(\tfrac{1}{\lambda n})\},$$

*where $b = \frac{1}{\langle x_i, x_i \rangle} \left( q^{\setminus y_i} + (1 - q_{y_i})\mathbf{1} \right)$, $q = W^\top x_i - \langle x_i, x_i \rangle a_i$ and $\rho = 1$.*

We discuss in the following section how to project onto the set $\Delta_k(\frac{1}{\lambda n})$ efficiently.

# 4 Efficient Projection onto the Top-$k$ Simplex

One of our main technical results is an algorithm for efficiently computing projections onto $\Delta_k(r)$, respectively the biased projection introduced in Proposition 4. The optimization problem in Proposition 4 reduces to the Euclidean projection onto $\Delta_k(r)$ for $\rho = 0$, and for $\rho > 0$ it biases the solution to be orthogonal to $\mathbf{1}$. Let us highlight that $\Delta_k(r)$ is substantially different from the standard simplex and none of the existing methods can be used as we discuss below.

## 4.1 Continuous Quadratic Knapsack Problem

Finding the Euclidean projection onto the simplex is an instance of the general optimization problem $\min_x \{\|a - x\|_2^2 \mid \langle b, x \rangle \leq r, \ l \leq x_i \leq u\}$ known as the *continuous quadratic knapsack problem* (CQKP). For example, to project onto the simplex we set $b = \mathbf{1}$, $l = 0$ and $r = u = 1$. This is a well examined problem and several highly efficient algorithms are available (see the surveys [18, 19]). The first main difference to our set is the upper bound on the $x_i$'s. All existing algorithms expect that $u$ is *fixed*, which allows them to consider decompositions $\min_{x_i} \{(a_i - x_i)^2 \mid l \leq x_i \leq u\}$ which can be solved in closed-form. In our case, the upper bound $\frac{1}{k} \langle \mathbf{1}, x \rangle$ introduces coupling across all variables, which makes the existing algorithms not applicable. A second main difference is the bias term $\rho \langle \mathbf{1}, x \rangle^2$ added to the objective. The additional difficulty introduced by this term is relatively minor. Thus we solve the problem for general $\rho$ (including $\rho = 0$ for the Euclidean projection onto $\Delta_k(r)$) even though we need only $\rho = 1$ in Proposition 4. The only case when our problem reduces to CQKP is when the constraint $\langle \mathbf{1}, x \rangle \leq r$ is satisfied with equality. In that case we can let $u = r/k$ and use any algorithm for the knapsack problem. We choose [13] since it is easy to implement, does not require sorting, and scales linearly in practice. The bias in the projection problem reduces to a constant $\rho r^2$ in this case and has, therefore, no effect.

## 4.2 Projection onto the Top-$k$ Cone

When the constraint $\langle \mathbf{1}, x \rangle \leq r$ is not satisfied with equality at the optimum, it has essentially no influence on the projection problem and can be removed. In that case we are left with the problem of the (biased) projection onto the top-$k$ cone which we address with the following lemma.

**Lemma 3.** *Let $x^* \in \mathbb{R}^d$ be the solution to the following optimization problem*

$$\min_x \{\|a - x\|^2 + \rho \langle \mathbf{1}, x \rangle^2 \mid 0 \leq x_i \leq \tfrac{1}{k} \langle \mathbf{1}, x \rangle, \ i \in [d]\},$$

*and let $U \triangleq \{i \mid x_i^* = \frac{1}{k} \langle \mathbf{1}, x^* \rangle\}$, $M \triangleq \{i \mid 0 < x_i^* < \frac{1}{k} \langle \mathbf{1}, x^* \rangle\}$, $L \triangleq \{i \mid x_i^* = 0\}$.*

1. *If $U = \varnothing$ and $M = \varnothing$, then $x^* = 0$.*

2. *If $U \neq \varnothing$ and $M = \varnothing$, then $U = \{[1], \ldots, [k]\}$, $x_i^* = \frac{1}{k + \rho k^2} \sum_{i=1}^{k} a_{[i]}$ for $i \in U$, where $[i]$ is the index of the $i$-th largest component in $a$.*

3. *Otherwise ($M \neq \varnothing$), the following system of linear equations holds*

$$\begin{cases} u & = \left(|M| \sum_{i \in U} a_i + (k - |U|) \sum_{i \in M} a_i\right)/D, \\ t' & = \left(|U|(1 + \rho k) \sum_{i \in M} a_i - (k - |U| + \rho k |M|) \sum_{i \in U} a_i\right)/D, \\ D & = (k - |U|)^2 + (|U| + \rho k^2) |M|, \end{cases} \quad (6)$$

   *together with the feasibility constraints on $t \triangleq t' + \rho u k$*

$$\max_{i \in L} a_i \leq t \leq \min_{i \in M} a_i, \qquad \max_{i \in M} a_i \leq t + u \leq \min_{i \in U} a_i, \qquad (7)$$

   *and we have $x^* = \min\{\max\{0, a - t\}, u\}$.*

We now show how to check if the (biased) projection is $0$. For the standard simplex, where the cone is the positive orthant $\mathbb{R}_+^d$, the projection is $0$ when all $a_i \leq 0$. It is slightly more involved for $\Delta_k$.

**Lemma 4.** *The biased projection $x^*$ onto the top-$k$ cone is zero if $\sum_{i=1}^{k} a_{[i]} \leq 0$ (sufficient condition). If $\rho = 0$ this is also necessary.*

**Projection.** Lemmas 3 and 4 suggest a simple algorithm for the (biased) projection onto the top-$k$ cone. First, we check if the projection is constant (cases 1 and 2 in Lemma 3). In case 2, we compute $x$ and check if it is compatible with the corresponding sets $U$, $M$, $L$. In the general case 3, we suggest a simple exhaustive search strategy. We sort $a$ and loop over the feasible partitions $U$, $M$, $L$ until we find a solution to (6) that satisfies (7). Since we know that $0 \leq |U| < k$ and $k \leq |U| + |M| \leq d$, we can limit the search to $(k-1)(d-k+1)$ iterations in the worst case, where each iteration requires a constant number of operations. For the biased projection, we leave $x = 0$ as the fallback case as Lemma 4 gives only a sufficient condition. This yields a runtime complexity of $O(d \log(d) + kd)$, which is comparable to simplex projection algorithms based on sorting.

### 4.3 Projection onto the Top-$k$ Simplex

As we argued in § 4.1, the (biased) projection onto the top-$k$ simplex becomes either the knapsack problem or the (biased) projection onto the top-$k$ cone depending on the constraint $\langle \mathbf{1}, x \rangle \leq r$ at the optimum. The following Lemma provides a way to check which of the two cases apply.

**Lemma 5.** *Let $x^* \in \mathbb{R}^d$ be the solution to the following optimization problem*

$$\min_x \{ \|a - x\|^2 + \rho \langle \mathbf{1}, x \rangle^2 \mid \langle \mathbf{1}, x \rangle \leq r, \ 0 \leq x_i \leq \tfrac{1}{k} \langle \mathbf{1}, x \rangle, \ i \in [d] \},$$

*let $(t, u)$ be the optimal thresholds such that $x^* = \min\{\max\{0, a - t\}, u\}$, and let $U$ be defined as in Lemma 3. Then it must hold that $\lambda = t + \frac{p}{k} - \rho r \geq 0$, where $p = \sum_{i \in U} a_i - |U|(t + u)$.*

**Projection.** We can now use Lemma 5 to compute the (biased) projection onto $\Delta_k(r)$ as follows. First, we check the special cases of zero and constant projections, as we did before. If that fails, we proceed with the knapsack problem since it is faster to solve. Having the thresholds $(t, u)$ and the partitioning into the sets $U$, $M$, $L$, we compute the value of $\lambda$ as given in Lemma 5. If $\lambda \geq 0$, we are done. Otherwise, we know that $\langle \mathbf{1}, x \rangle < r$ and go directly to the general case 3 in Lemma 3.

## 5 Experimental Results

We have two main goals in the experiments. First, we show that the (biased) projection onto the top-$k$ simplex is scalable and comparable to an efficient algorithm [13] for the simplex projection (see the supplement). Second, we show that the top-$k$ multiclass SVM using both versions of the top-$k$ hinge loss (3) and (5), denoted $\text{top-}k \text{ SVM}_\alpha$ and $\text{top-}k \text{ SVM}_\beta$ respectively, leads to improvements in top-$k$ accuracy consistently over all datasets and choices of $k$. In particular, we note improvements compared to the multiclass SVM of Crammer and Singer [5], which corresponds to $\text{top-1 SVM}_\alpha / \text{top-1 SVM}_\beta$. We release our implementation of the projection procedures and both SDCA solvers as a C++ library[2] with a Matlab interface.

### 5.1 Image Classification Experiments

We evaluate our method on five image classification datasets of different scale and complexity: Caltech 101 Silhouettes [26] ($m = 101$, $n = 4100$), MIT Indoor 67 [20] ($m = 67$, $n = 5354$), SUN 397 [29] ($m = 397$, $n = 19850$), Places 205 [30] ($m = 205$, $n = 2448873$), and ImageNet 2012 [22] ($m = 1000$, $n = 1281167$). For Caltech, $d = 784$, and for the others $d = 4096$. The results on the two large scale datasets are in the supplement.

We cross-validate hyper-parameters in the range $10^{-5}$ to $10^3$, extending it when the optimal value is at the boundary. We use LibLinear [7] for $\text{SVM}^{\text{OVA}}$, $\text{SVM}^{\text{Perf}}$ [11] with the corresponding loss function for $\text{Recall@k}$, and the code provided by [16] for $\text{TopPush}$. When a ranking method like $\text{Recall@k}$ and $\text{TopPush}$ does not scale to a particular dataset using the reduction of the multiclass to a binary problem discussed in § 2.3, we use the one-vs-all version of the corresponding method. We implemented $\text{Wsabie}^{++}$ (denoted $\text{W}_{++}$, $\text{Q/m}$) based on the pseudo-code from Table 3 in [9].

On Caltech 101, we use features provided by [26]. For the other datasets, we extract CNN features of a pre-trained CNN (fc7 layer after ReLU). For the scene recognition datasets, we use the Places 205 CNN [30] and for ILSVRC 2012 we use the Caffe reference model [10].

**Caltech 101 Silhouettes** / **MIT Indoor 67**

| Method | Top-1 | Top-2 | Top-3 | Top-4 | Top-5 | Top-10 | Method | Top-1 | Method | Top-1 | Method | Top-1 |
|---|---|---|---|---|---|---|---|---|---|---|---|---|
| Top-1 [26] | 62.1 | - | 79.6 | - | 83.1 | - | BLH [4] | 48.3 | DGE [6] | 66.87 | RAS [21] | 69.0 |
| Top-2 [26] | 61.4 | - | 79.2 | - | 83.4 | - | SP [25] | 51.4 | ZLX [30] | 68.24 | KL [14] | 70.1 |
| Top-5 [26] | 60.2 | - | 78.7 | - | 83.4 | - | JVJ [12] | 63.10 | GWG [8] | 68.88 | | |

| Method | Top-1 | Top-2 | Top-3 | Top-4 | Top-5 | Top-10 | Top-1 | Top-2 | Top-3 | Top-4 | Top-5 | Top-10 |
|---|---|---|---|---|---|---|---|---|---|---|---|---|
| SVM$^{\text{OVA}}$ | 61.81 | 73.13 | 76.25 | 77.76 | 78.89 | 83.57 | 71.72 | 81.49 | 84.93 | 86.49 | 87.39 | 90.45 |
| TopPush | 63.11 | 75.16 | 78.46 | 80.19 | 81.97 | 86.95 | 70.52 | 83.13 | 86.94 | 90.00 | 91.64 | 95.90 |
| Recall@1 | 61.55 | 73.13 | 77.03 | 79.41 | 80.97 | 85.18 | 71.57 | 83.06 | 87.69 | 90.45 | 92.24 | 96.19 |
| Recall@5 | 61.60 | 72.87 | 76.51 | 78.76 | 80.54 | 84.74 | 71.49 | 81.49 | 85.45 | 87.24 | 88.21 | 92.01 |
| Recall@10 | 61.51 | 72.95 | 76.46 | 78.72 | 80.54 | 84.92 | 71.42 | 81.49 | 85.52 | 87.24 | 88.28 | 92.16 |
| $W_{++,\,0/256}$ | 62.68 | 76.33 | 79.41 | 81.71 | 83.18 | 88.95 | 70.07 | 84.10 | 89.48 | 92.46 | 94.48 | **97.91** |
| $W_{++,\,1/256}$ | 59.25 | 65.63 | 69.22 | 71.09 | 72.95 | 79.71 | 68.13 | 81.49 | 86.64 | 89.63 | 91.42 | 95.45 |
| $W_{++,\,2/256}$ | 55.09 | 61.81 | 66.02 | 68.88 | 70.61 | 76.59 | 64.63 | 78.43 | 84.18 | 88.13 | 89.93 | 94.55 |
| top-1 SVM$_\alpha$ | 62.81 | 74.60 | 77.76 | 80.02 | 81.97 | 86.91 | **73.96** | 85.22 | 89.25 | 91.94 | 93.43 | 96.94 |
| top-10 SVM$_\alpha$ | 62.98 | **77.33** | 80.49 | 82.66 | 84.57 | 89.55 | 70.00 | **85.45** | 90.00 | 93.13 | **94.63** | 97.76 |
| top-20 SVM$_\alpha$ | 59.21 | 75.64 | 80.88 | **83.49** | **85.39** | **90.33** | 65.90 | 84.10 | 89.93 | 92.69 | 94.25 | 97.54 |
| top-1 SVM$_\beta$ | 62.81 | 74.60 | 77.76 | 80.02 | 81.97 | 86.91 | **73.96** | 85.22 | 89.25 | 91.94 | 93.43 | 96.94 |
| top-10 SVM$_\beta$ | **64.02** | 77.11 | 80.49 | 83.01 | 84.87 | 89.42 | 71.87 | 85.30 | **90.45** | **93.36** | 94.40 | 97.76 |
| top-20 SVM$_\beta$ | 63.37 | 77.24 | **81.06** | 83.31 | 85.18 | 90.03 | 71.94 | 85.30 | 90.07 | 92.46 | 94.33 | 97.39 |

**SUN 397** (10 splits)

| Top-1 accuracy | XHE [29] 38.0 | LSH [15] 49.48 ± 0.3 | ZLX [30] 54.32 ± 0.1 |
|---|---|---|---|
| | SPM [23] 47.2 ± 0.2 | GWG [8] 51.98 | KL [14] 54.65 ± 0.2 |

| Method | Top-1 | Top-2 | Top-3 | Top-4 | Top-5 | Top-10 |
|---|---|---|---|---|---|---|
| SVM$^{\text{OVA}}$ | 55.23 ± 0.6 | 66.23 ± 0.6 | 70.81 ± 0.4 | 73.30 ± 0.2 | 74.93 ± 0.2 | 79.00 ± 0.3 |
| TopPush$^{\text{OVA}}$ | 53.53 ± 0.3 | 65.39 ± 0.3 | 71.46 ± 0.2 | 75.25 ± 0.1 | 77.95 ± 0.2 | 85.15 ± 0.3 |
| Recall@1$^{\text{OVA}}$ | 52.95 ± 0.2 | 65.49 ± 0.2 | 71.86 ± 0.2 | 75.88 ± 0.2 | 78.72 ± 0.2 | 86.03 ± 0.2 |
| Recall@5$^{\text{OVA}}$ | 50.72 ± 0.2 | 64.74 ± 0.3 | 70.75 ± 0.3 | 74.02 ± 0.3 | 76.06 ± 0.3 | 80.66 ± 0.2 |
| Recall@10$^{\text{OVA}}$ | 50.92 ± 0.2 | 64.94 ± 0.2 | 70.95 ± 0.2 | 74.14 ± 0.2 | 76.21 ± 0.2 | 80.68 ± 0.2 |
| top-1 SVM$_\alpha$ | 58.16 ± 0.2 | 71.66 ± 0.2 | 78.22 ± 0.1 | 82.29 ± 0.2 | 84.98 ± 0.2 | 91.48 ± 0.2 |
| top-10 SVM$_\alpha$ | 58.00 ± 0.2 | 73.65 ± 0.1 | 80.80 ± 0.1 | 84.81 ± 0.2 | 87.45 ± 0.2 | 93.40 ± 0.2 |
| top-20 SVM$_\alpha$ | 55.98 ± 0.3 | 72.51 ± 0.2 | 80.22 ± 0.2 | 84.54 ± 0.2 | 87.37 ± 0.2 | 93.62 ± 0.2 |
| top-1 SVM$_\beta$ | 58.16 ± 0.2 | 71.66 ± 0.2 | 78.22 ± 0.1 | 82.29 ± 0.2 | 84.98 ± 0.2 | 91.48 ± 0.2 |
| top-10 SVM$_\beta$ | **59.32 ± 0.1** | **74.13 ± 0.2** | 80.91 ± 0.2 | 84.92 ± 0.2 | 87.49 ± 0.2 | 93.36 ± 0.2 |
| top-20 SVM$_\beta$ | 58.65 ± 0.2 | 73.96 ± 0.2 | **80.95 ± 0.2** | **85.05 ± 0.2** | **87.70 ± 0.2** | **93.64 ± 0.2** |

Table 1: Top-$k$ accuracy (%). **Top section:** State of the art. **Middle section:** Baseline methods. **Bottom section:** Top-$k$ SVMs: top-$k$ SVM$_\alpha$ – with the loss (3); top-$k$ SVM$_\beta$ – with the loss (5).

Experimental results are given in Table 1. First, we note that our method is scalable to large datasets with millions of training examples, such as Places 205 and ILSVRC 2012 (results in the supplement). Second, we observe that optimizing the top-$k$ hinge loss (both versions) yields consistently better top-$k$ performance. This might come at the cost of a decreased top-1 accuracy (e.g. on MIT Indoor 67), but, interestingly, may also result in a noticeable increase in the top-1 accuracy on larger datasets like Caltech 101 Silhouettes and SUN 397. This resonates with our argumentation that optimizing for top-$k$ is often more appropriate for datasets with a large number of classes.

Overall, we get systematic increase in top-$k$ accuracy over all datasets that we examined. For example, we get the following improvements in top-5 accuracy with our top-10 SVM$_\alpha$ compared to top-1 SVM$_\alpha$: +2.6% on Caltech 101, +1.2% on MIT Indoor 67, and +2.5% on SUN 397.

## 6 Conclusion

We demonstrated scalability and effectiveness of the proposed top-$k$ multiclass SVM on five image recognition datasets leading to consistent improvements in top-$k$ performance. In the future, one could study if the top-$k$ hinge loss (3) can be generalized to the family of ranking losses [27]. Similar to the top-$k$ loss, this could lead to tighter convex upper bounds on the corresponding discrete losses.

## Footnotes

[1] A convex function $f : X \to \mathbb{R} \cup \{\pm\infty\}$ has an *effective domain* $\operatorname{dom} f = \{x \in X \mid f(x) < +\infty\}$.

[2]https://github.com/mlapin/libsdca

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
