[Supplementary Material · paper_195_supplementary.pdf]

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

\} \leq kt_0 + \sum_{j=1}^{m} [h_j - t_0]_+ = kt_0 + \sum_{j=1}^{k} \left( h_{[j]} - t_0 \right) = \sum_{i=1}^{k} h_{[i]}.$$

On the other hand, for any $t \in \mathbb{R}$, we get

$$\sum_{j=1}^{k} h_{[j]} = kt + \sum_{j=1}^{k} \left( h_{[j]} - t \right) \leq kt + \sum_{j=1}^{k} \left[ h_{[j]} - t \right]_+ \leq kt + \sum_{j=1}^{m} [h_j - t]_+ .$$

$\square$

Figure 2: Top-$k$ simplex $\Delta_k(1)$ for $m = 3$. Unlike the standard simplex, it has $\binom{m}{k} + 1$ vertices.

We also define a set $\Delta_k$ which arises naturally as the effective domain[1] of the conjugate of (3). By analogy, we call it the top-$k$ simplex as for $k = 1$ it reduces to the standard simplex with the inequality constraint (i.e. $0 \in \Delta_k$). Let $[m] \triangleq 1, \dots, m$.

**Definition 1.** *The* top-$k$ simplex *is a convex polytope defined as*

$$\Delta_k(r) \triangleq \left\{ x \,\middle|\, \langle \mathbf{1}, x \rangle \leq r, \ 0 \leq x_i \leq \frac{1}{k} \langle \mathbf{1}, x \rangle, \ i \in [m] \right\},$$

*where $r \geq 0$ is the bound on the sum $\langle \mathbf{1}, x \rangle$. We let $\Delta_k \triangleq \Delta_k(1)$.*

The crucial difference to the standard simplex is the upper bound on $x_i$'s, which limits their maximal contribution to the total sum $\langle \mathbf{1}, x \rangle$. See Figure 2 for an illustration.

The first technical contribution of this work is as follows.

**Proposition 2.** *A primal-conjugate pair for the top-$k$ hinge loss (3) is given as follows:*

$$\phi_k(a) = \max \left\{ 0, \frac{1}{k} \sum_{j=1}^{k} (a+c)_{[j]} \right\}, \qquad \phi_k^*(b) = \begin{cases} -\langle c, b \rangle & \text{if } b \in \Delta_k, \\ +\infty & \text{otherwise.} \end{cases} \tag{4}$$

*Moreover, $\phi_k(a) = \max\{\langle a + c, \lambda \rangle \mid \lambda \in \Delta_k\}$.*

*Proof.* We use Lemma 1 to write

$$\phi_k(a) = \min\left\{s \mid s \geq t + \frac{1}{k}\sum_{j=1}^{m}\xi_j,\ s \geq 0,\ \xi_j \geq a_j + c_j - t,\ \xi_j \geq 0\right\}.$$

The Lagrangian is given as

$$\mathcal{L}(s,t,\xi,\alpha,\beta,\lambda,\mu) = s + \alpha\left(t + \frac{1}{k}\sum_{j=1}^{m}\xi_j - s\right) - \beta s + \sum_{j=1}^{m}\lambda_j\left(a_j + c_j - t - \xi_j\right) - \sum_{j=1}^{m}\mu_j\xi_j.$$

Minimizing over $(s,t,\xi)$, we get $\alpha + \beta = 1$, $\alpha = \sum_{j=1}^{m}\lambda_j$, $\lambda_j + \mu_j = \frac{1}{k}\alpha$. As $\beta \geq 0$ and $\mu_j \geq 0$, it follows that $\langle \mathbf{1}, \lambda \rangle \leq 1$ and $0 \leq \lambda_j \leq \frac{1}{k}\langle \mathbf{1}, \lambda \rangle$. Since the duality gap is zero, we get

$$\phi_k(a) = \max\{\langle a + c, \lambda \rangle \mid \lambda \in \Delta_k\}.$$

The conjugate $\phi_k^*(b)$ can now be computed as

$$\max_a\{\langle a, b \rangle - \phi_k(a)\} = \max_a \min_{\lambda \in \Delta_k}\{\langle a, b \rangle - \langle a + c, \lambda \rangle\} = \min_{\lambda \in \Delta_k}\{-\langle c, \lambda \rangle + \max_a \langle a, b - \lambda \rangle\}.$$

Since $\max_a \langle a, b - \lambda \rangle = \infty$ unless $b = \lambda$, we get the formula for $\phi_k^*(b)$ as in (4).

$\square$

Therefore, we see that the proposed formulation (3) naturally extends the multiclass SVM of Crammer and Singer [6], which is recovered when $k = 1$. We have also obtained an interesting extension (or rather contraction, since $\Delta_k \subset \Delta$) of the standard simplex.

## 2.3 Relation of the Top-$k$ Hinge Loss to Ranking Based Losses

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

*Proof.* We have that $\operatorname{Ker} H_j = \{x \,|\, H_j x = 0\} = \{t\mathbf{1} \,|\, t \in \mathbb{R}\}$ and $\operatorname{Ker}^\perp H_j = \{x \,|\, \langle \mathbf{1}, x \rangle = 0\}$.

$$\begin{aligned}
\Phi^*(y) &= \sup\{\langle y, x \rangle - \Phi(x) \,|\, x \in \mathbb{R}^m\} \\
&= \sup\{\langle y, x^\| \rangle + \langle y, x^\perp \rangle - \phi(H_j x^\perp) \,|\, x = x^\| + x^\perp, x^\| \in \operatorname{Ker} H_j, x^\perp \in \operatorname{Ker}^\perp H_j\}.
\end{aligned}$$

It follows that $\Phi^*(y)$ can only be finite if $\langle y, x^\| \rangle = 0$, which implies $y \in \operatorname{Ker}^\perp H_j$. Let $H_j^\dagger$ be the Moore-Penrose pseudoinverse of $H_j$. For a $y \in \operatorname{Ker}^\perp H_j$, we can write

$$\begin{aligned}
\Phi^*(y) &= \sup\{\langle y, H_j^\dagger H_j x^\perp \rangle - \phi(H_j x^\perp) \,|\, x^\perp \in \operatorname{Ker}^\perp H_j\} \\
&= \sup\{\langle (H_j^\dagger)^\top y, z \rangle - \phi(z) \,|\, z \in \operatorname{Im} H_j\} \qquad\qquad (6) \\
&\le \sup\{\langle (H_j^\dagger)^\top y, z \rangle - \phi(z) \,|\, z \in \mathbb{R}^m\} = \phi^*((H_j^\dagger)^\top y),
\end{aligned}$$

where $\operatorname{Im} H_j = \{H_j x \,|\, x \in \mathbb{R}^m\}$. Using rank-1 update of the Moore-Penrose pseudoinverse ([22], § 3.2.7), we can compute $(H_j^\dagger)^\top = \mathbf{I} - e_j e_j^\top - \frac{1}{m}(\mathbf{1} - e_j)\mathbf{1}^\top$. Since $y \in \operatorname{Ker}^\perp H_j$, the last term is zero and we have $(H_j^\dagger)^\top y = y - y_j e_j$. Finally, we use the fact that $\phi$ is $j$-compatible to prove that the inequality in (6) is satisfied with equality. We have that $\operatorname{Im} H_j = \{z \,|\, z_j = 0\}$ and $(y - y_j e_j)_j = 0$. Therefore, when $\langle \mathbf{1}, y \rangle = 0$, $\Phi^*(y) = \sup\{\langle y - y_j e_j, z \rangle - \phi(z) \,|\, z_j = 0\} = \phi^*(y - y_j e_j)$. $\qquad\square$

We can now use Lemma 2 to compute convex conjugates of the loss functions.

**Theorem 1.** *Let $\phi_i$ be $y_i$-compatible for each $i \in [n]$, let $\lambda > 0$ be a regularization parameter, and let $K = X^\top X$ be the Gram matrix. The primal and Fenchel dual objective functions are given as:*

$$P(W) = +\frac{1}{n}\sum_{i=1}^n \phi_i\left(W^\top x_i - \langle w_{y_i}, x_i \rangle \mathbf{1}\right) + \frac{\lambda}{2}\operatorname{tr}\left(W^\top W\right),$$

$$D(A) = -\frac{1}{n}\sum_{i=1}^n \phi_i^*\left(-\lambda n(a_i - a_{y_i,i}e_{y_i})\right) - \frac{\lambda}{2}\operatorname{tr}\left(AKA^\top\right), \text{ if } \langle \mathbf{1}, a_i \rangle = 0 \; \forall i, \; +\infty \text{ otherwise.}$$

*Moreover, we have that $W = XA^\top$ and $W^\top x_i = AK_i$, where $K_i$ is the $i$-th column of $K$.*

*Proof.* We use Fenchel duality (see e.g. [2], Theorem 3.3.5), to write $P(W) = g(X^\top W) + f(W)$, and $D(A) = -g^*(-A^\top) - f^*(XA^\top)$, for the functions $g$ and $f$ defined as follows:

$$g(X^\top W) = \frac{1}{n} \sum_{i=1}^{n} \Phi_i \left( W^\top x_i \right) = \frac{1}{n} \sum_{i=1}^{n} \phi_i \left( H_{y_i} W^\top x_i \right), \quad f(W) = \frac{\lambda}{2} \operatorname{tr} \left( W^\top W \right) = \frac{\lambda}{2} \|W\|_F^2,$$

where $H_{y_i} = \mathbf{I} - \mathbf{1} e_{y_i}^\top$. One can easily verify that $g^*(-A^\top) = \frac{1}{n} \sum_{i=1}^{n} \Phi_i^*(-na_i)$ and $f^*(XA^\top) = \frac{\lambda}{2} \left\| \frac{1}{\lambda} XA^\top \right\|_F^2$. From Lemma 2, we have that $\Phi_i^*(-na_i) = \phi^*(-n(a_i - a_{y_i,i} e_{y_i}))$, if $\langle \mathbf{1}, -na_i \rangle = 0$, and $+\infty$ otherwise. To complete the proof, we redefine $A \leftarrow \frac{1}{\lambda} A$ for convenience, and use the first order optimality condition ([2], Ex. 9.f in § 3) for the $W = XA^\top$ formula. $\qquad \square$

Finally, we show that Theorem 1 applies to the loss functions that we consider.

**Proposition 3.** *The top-$k$ hinge loss function from Section 2 is $y_i$-compatible.*

*Proof.* Let $c = \mathbf{1} - e_{y_i}$ and consider the loss $\phi_k$. As in Proposition 2, we have

$$\max_{a,\, a_{y_i}=0} \{ \langle a, b \rangle - \phi_k(a) \} = \min_{\lambda \in \Delta_k} \left\{ -\langle c, \lambda \rangle + \max_{a,\, a_{y_i}=0} \langle a, b - \lambda \rangle \right\} = \phi_k^*(b),$$

where we used that $c_{y_i} = 0$ and $b_{y_i} = 0$ (cf. Definition 2), i.e. the $y_i$-th coordinate has no influence. $\qquad \square$

We have repeated the derivation from Section 5.7 in [27] as there is a typo in the optimization problem (20) leading to the conclusion that $a_{y_i,i}$ must be 0 at the optimum. Lemma 2 fixes this by making the requirement $a_{y_i,i} = -\sum_{j \neq y_i} a_{j,i}$ explicit. Note that this modification is already mentioned in their pseudo-code for Prox-SDCA.

### 3.2 Optimization of Top-$k$ Multiclass SVM via Prox-SDCA

As an optimization scheme, we employ the proximal stochastic dual coordinate ascent (Prox-SDCA) framework of Shalev-Shwartz and Zhang [27], which has strong convergence guarantees and is easy to adapt to our problem. In particular, we iteratively update a batch $a_i \in \mathbb{R}^m$ of dual variables corresponding to the training pair $(x_i, y_i)$, so as to maximize the dual objective $D(A)$ from Theorem 1. We also maintain the primal variables $W = XA^\top$ and stop when the relative duality gap is below $\epsilon$. This procedure is summarized in Algorithm 1.

Let us make a few comments on the advantages of the proposed method. First, apart from the update step which we discuss below, all main operations can be computed using a BLAS library, which makes the overall implementation efficient. Second, the update step in Line 9 is optimal in the sense that it yields maximal dual objective increase jointly over $m$ variables. This is opposed to SGD updates with data-independent step sizes, as well as to maximal but *scalar* updates in other SDCA variants. Finally, we have a well-defined stopping criterion as we can compute the duality gap (see discussion in [3]). The latter is especially attractive if there is a time budget for learning. The algorithm can also be easily kernelized since $W^\top x_i = AK_i$ (cf. Theorem 1).

---

**Algorithm 1** Top-$k$ Multiclass SVM

1: **Input:** training data $\{(x_i, y_i)_{i=1}^n\}$, parameters $k$ (loss), $\lambda$ (regularization), $\epsilon$ (stopping cond.)
2: **Output:** $W \in \mathbb{R}^{d \times m}$, $A \in \mathbb{R}^{m \times n}$
3: **Initialize:** $W \leftarrow 0$, $A \leftarrow 0$
4: **repeat**
5:     randomly permute training data
6:     **for** $i = 1$ **to** $n$ **do**
7:         $s_i \leftarrow W^\top x_i$ {prediction scores}
8:         $a_i^{\mathrm{old}} \leftarrow a_i$ {cache previous values}
9:         $a_i \leftarrow update(k, \lambda, \|x_i\|^2, y_i, s_i, a_i)$
        {see § 3.2.1 for details}
10:      $W \leftarrow W + x_i(a_i - a_i^{\mathrm{old}})^\top$
        {rank-1 update}
11:     **end for**
12: **until** relative duality gap is below $\epsilon$

---

#### 3.2.1 Dual Variables Update

For the proposed top-$k$ hinge loss from Section 2, optimization of the dual objective $D(A)$ over $a_i \in \mathbb{R}^m$ given other variables fixed is an instance of a regularized (biased) projection problem onto the top-$k$ simplex $\Delta_k(\frac{1}{\lambda n})$. Let $a^{\backslash j}$ be obtained by removing the $j$-th coordinate from vector $a$.

**Proposition 4.** *The following two problems are equivalent with $a_i^{\backslash y_i} = -x$ and $a_{y_i,i} = \langle \mathbf{1}, x \rangle$*

$$\max_{a_i}\{D(A) \mid \langle \mathbf{1}, a_i \rangle = 0\} \equiv \min_x\{\|b - x\|^2 + \rho \langle \mathbf{1}, x \rangle^2 \mid x \in \Delta_k(\tfrac{1}{\lambda n})\},$$

*where $b = \frac{1}{\langle x_i, x_i \rangle}\left(q^{\backslash y_i} + (1 - q_{y_i})\mathbf{1}\right)$, $q = W^\top x_i - \langle x_i, x_i \rangle a_i$ and $\rho = 1$.*

*Proof.* Using Proposition 2 and Theorem 1, we write

$$\max_{a_i}\{ -\frac{1}{n}\phi_i^*\left(-\lambda n(a_i - a_{y_i,i}e_{y_i})\right) - \frac{\lambda}{2}\operatorname{tr}\left(AKA^\top\right) \mid \langle \mathbf{1}, a_i \rangle = 0\}.$$

For the loss function, we get

$$-\frac{1}{n}\phi_i^*\left(-\lambda n(a_i - a_{y_i,i}e_{y_i})\right) = \lambda a_{y_i,i},$$

with $-\lambda n(a_i - a_{y_i,i}e_{y_i}) \in \Delta_k$. One can verify that the latter constraint is equivalent to $-a_i^{\backslash y_i} \in \Delta_k(\tfrac{1}{\lambda n})$, $a_{y_i,i} = \langle \mathbf{1}, -a_i^{\backslash y_i}\rangle$. Similarly, we write for the regularization term

$$\operatorname{tr}\left(AKA^\top\right) = K_{ii}\langle a_i, a_i \rangle + 2\sum_{j \neq i} K_{ij}\langle a_i, a_j \rangle + \text{const},$$

where the $\operatorname{const}$ does not depend on $a_i$. Note that $\sum_{j \neq i} K_{ij}a_j = AK_i - K_{ii}a_i = q$ and can be computed using the "old" $a_i$. Let $x \triangleq -a_i^{\backslash y_i}$, we have

$$\langle a_i, a_i \rangle = \langle \mathbf{1}, x \rangle^2 + \langle x, x \rangle, \qquad\qquad \langle q, a_i \rangle = q_{y_i}\langle \mathbf{1}, x \rangle - \langle q^{\backslash y_i}, x \rangle.$$

Plugging everything together and multiplying with $-2/\lambda$, we obtain

$$\min_{x \in \Delta_k(\tfrac{1}{\lambda n})} -2\langle \mathbf{1}, x \rangle + 2\left(q_{y_i}\langle \mathbf{1}, x \rangle - \langle q^{\backslash y_i}, x \rangle\right) + K_{ii}\left(\langle \mathbf{1}, x \rangle^2 + \langle x, x \rangle\right).$$

Collecting the corresponding terms finishes the proof. $\qquad\square$

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

*and we have $x^* = \min\{\max\{0, a - t\}, u\}$.*

*Proof.* We consider an equivalent problem

$$\min_{x,s} \{ \tfrac{1}{2} \|a - x\|^2 + \tfrac{1}{2} \rho s^2 \mid \langle \mathbf{1}, x \rangle = s, \ 0 \leq x_i \leq \tfrac{s}{k}, \ i \in [d] \}.$$

Let $t$, $\mu_i \geq 0$, $\nu_i \geq 0$ be the dual variables, and let $\mathcal{L}$ be the Lagrangian:

$$\mathcal{L}(x, s, t, \mu, \nu) = \tfrac{1}{2} \|a - x\|^2 + \tfrac{1}{2} \rho s^2 + t(\langle \mathbf{1}, x \rangle - s) - \langle \mu, x \rangle + \langle \nu, x - \tfrac{s}{k} \mathbf{1} \rangle.$$

From the KKT conditions, we have that

$$\partial_x \mathcal{L} = x - a + t\mathbf{1} - \mu + \nu = 0, \quad \partial_s \mathcal{L} = \rho s - t - \tfrac{1}{k} \langle \mathbf{1}, \nu \rangle = 0, \quad \mu_i x_i = 0, \quad \nu_i (x_i - \tfrac{s}{k}) = 0.$$

We have that $x_i = \min\{\max\{0, a_i - t\}, \tfrac{s}{k}\}$, $\nu_i = \max\{0, a_i - t - \tfrac{s}{k}\}$, and $s = \tfrac{1}{\rho}(t + \tfrac{1}{k} \langle \mathbf{1}, \nu \rangle)$. Let $p \triangleq \langle \mathbf{1}, \nu \rangle$. We have $t = \rho s - \tfrac{p}{k}$. Using the definition of the sets $U$ and $M$, we get

$$s = \sum_{i \in U} \frac{s}{k} + \sum_{i \in M} (a_i - t) = \sum_{i \in M} a_i - |M| \left( \rho s - \frac{p}{k} \right) + |U| \frac{s}{k},$$

$$p = \sum_{i \in U} (a_i - t - \frac{s}{k}) = \sum_{i \in U} a_i - |U| \left( \rho s - \frac{p}{k} \right) - |U| \frac{s}{k}.$$

In the case $U \neq \varnothing$ and $M = \varnothing$ we get the simplified equations

$$s = \sum_{i \in U} \frac{s}{k} = |U| \frac{s}{k} \quad \Longrightarrow \quad |U| = k,$$

$$p = \sum_{i \in U} a_i - k \rho s + p - s \quad \Longrightarrow \quad x_i = \frac{s}{k} = \frac{1}{k + \rho k^2} \sum_{i \in U} a_i, \ i \in U.$$

In the remaining case solving this system for $u \triangleq \tfrac{s}{k}$ and $t' \triangleq -\tfrac{p}{k}$, we get exactly the system in (7). The constraints (8) follow from the definition of the sets $U$, $M$, $L$, and ensure that the computed thresholds $(t, u)$ are compatible with the corresponding partitioning of the index set. $\qquad \square$

We now show how to check if the (biased) projection is $0$. For the standard simplex, where the cone is the positive orthant $\mathbb{R}_+^d$, the projection is $0$ when all $a_i \leq 0$. It is slightly more involved for $\Delta_k$.

**Lemma 4.** *The biased projection $x^*$ onto the top-$k$ cone is zero if $\sum_{i=1}^{k} a_{[i]} \leq 0$ (sufficient condition). If $\rho = 0$ this is also necessary.*

*Proof.* Let $K \triangleq \{x \mid 0 \leq x_i \leq \frac{1}{k} \langle \mathbf{1}, x \rangle\}$ be the top-$k$ cone. It is known that the Euclidean projection of $a$ onto $K$ is 0 if and only if $a \in N_K(0) \triangleq \{y \mid \forall x \in K, \langle y, x \rangle \leq 0\}$, i.e. $a$ is in the normal cone to $K$ at 0. Therefore, we obtain as an equivalent condition that $\max_{x \in K} \langle a, x \rangle \leq 0$. Take any $x \in K$ and let $s = \langle \mathbf{1}, x \rangle$. If $s > 0$, we have that at least $k$ components in $x$ must be positive. To maximize $\langle a, x \rangle$, we would have exactly $k$ positive $x_i = \frac{s}{k}$ corresponding to the $k$ largest components in $a$. That would result in $\langle a, x \rangle = \frac{s}{k} \sum_{i=1}^{k} a_{[i]}$, which is non-positive if and only if $\sum_{i=1}^{k} a_{[i]} \leq 0$.

For $\rho > 0$, the objective function has an additional term $\rho \langle \mathbf{1}, x \rangle^2$ that vanishes at $x = 0$. Therefore, if $x = 0$ is optimal for the Euclidean projection, it must also be optimal for the biased projection. $\square$

**Projection.** Lemmas 3 and 4 suggest a simple algorithm for the (biased) projection onto the top-$k$ cone. First, we check if the projection is constant (cases 1 and 2 in Lemma 3). In case 2, we compute $x$ and check if it is compatible with the corresponding sets $U, M, L$. In the general case 3, we suggest a simple exhaustive search strategy. We sort $a$ and loop over the feasible partitions $U, M, L$ until we find a solution to (7) that satisfies (8). Since we know that $0 \leq |U| < k$ and $k \leq |U| + |M| \leq d$, we can limit the search to $(k-1)(d-k+1)$ iterations in the worst case, where each iteration requires a constant number of operations. For the biased projection, we leave $x = 0$ as the fallback case as Lemma 4 gives only a sufficient condition. This yields a runtime complexity of $O(d \log(d) + kd)$, which is comparable to simplex projection algorithms based on sorting.

### 4.3 Projection onto the Top-$k$ Simplex

As we argued in § 4.1, the (biased) projection onto the top-$k$ simplex becomes either the knapsack problem or the (biased) projection onto the top-$k$ cone depending on the constraint $\langle \mathbf{1}, x \rangle \leq r$ at the optimum. The following Lemma provides a way to check which of the two cases apply.

**Lemma 5.** *Let $x^* \in \mathbb{R}^d$ be the solution to the following optimization problem*

$$\min_{x}\{\|a - x\|^2 + \rho \langle \mathbf{1}, x \rangle^2 \mid \langle \mathbf{1}, x \rangle \leq r, \ 0 \leq x_i \leq \tfrac{1}{k} \langle \mathbf{1}, x \rangle, \ i \in [d]\},$$

*let $(t, u)$ be the optimal thresholds such that $x^* = \min\{\max\{0, a - t\}, u\}$, and let $U$ be defined as in Lemma 3. Then it must hold that $\lambda = t + \frac{p}{k} - \rho r \geq 0$, where $p = \sum_{i \in U} a_i - |U| (t + u)$.*

*Proof.* As in Lemma 3, we consider an equivalent problem

$$\min_{x,s}\{\tfrac{1}{2} \|a - x\|^2 + \tfrac{1}{2} \rho s^2 \mid \langle \mathbf{1}, x \rangle = s, \ s \leq r, \ 0 \leq x_i \leq \tfrac{s}{k}, \ i \in [d]\}.$$

Let $t, \lambda \geq 0$, $\mu_i \geq 0$, $\nu_i \geq 0$ be the dual variables, and let $\mathcal{L}$ be the Lagrangian:

$$\mathcal{L} = \tfrac{1}{2} \|a - x\|^2 + \tfrac{1}{2} \rho s^2 + t(\langle \mathbf{1}, x \rangle - s) + \lambda(s - r) - \langle \mu, x \rangle + \langle \nu, x - \tfrac{s}{k} \mathbf{1} \rangle.$$

From the KKT conditions, we have that

$$\partial_x \mathcal{L} = x - a + t\mathbf{1} - \mu + \nu = 0, \qquad \partial_s \mathcal{L} = \rho s - t + \lambda - \tfrac{1}{k} \langle \mathbf{1}, \nu \rangle = 0,$$

$$\mu_i x_i = 0, \qquad\qquad \nu_i(x_i - \tfrac{s}{k}) = 0, \qquad \lambda(s - r) = 0.$$

If $s < r$, then $\lambda = 0$ and we recover the top-$k$ cone problem of Lemma 3. Otherwise, we have that $s = r$ and $\lambda = t + \frac{1}{k} \langle \mathbf{1}, \nu \rangle - \rho r \geq 0$. The fact that $\nu_i = \max\{0, a_i - t - u\}$, where $u = \frac{r}{k}$, completes the proof. $\square$

**Projection.** We can now use Lemma 5 to compute the (biased) projection onto $\Delta_k(r)$ as follows. First, we check the special cases of zero and constant projections, as we did before. If that fails, we proceed with the knapsack problem since it is faster to solve. Having the thresholds $(t, u)$ and the partitioning into the sets $U, M, L$, we compute the value of $\lambda$ as given in Lemma 5. If $\lambda \geq 0$, we are done. Otherwise, we know that $\langle \mathbf{1}, x \rangle < r$ and go directly to the general case 3 in Lemma 3.

# 5 Optimization of Top-$k$ Usunier Loss

In this section we show how the Usunier version of the top-$k$ hinge loss (5) can be optimized using the Prox-SDCA framework from § 3. The two main ingredients that we discuss are the conjugate loss and the (biased) projection. It turns out that the only difference between the conjugate of the top-$k$ hinge loss (3) introduced above and the conjugate of (5) are their effective domains.

**Proposition 5.** *A primal-conjugate pair for the top-k Usunier loss (5) is*

$$\tilde{\phi}_k(a) = \frac{1}{k} \sum_{j=1}^{k} \max\left\{0, (a+c)_{[j]}\right\}, \qquad \tilde{\phi}_k^*(b) = \begin{cases} -\langle c, b \rangle & \text{if } b \in \tilde{\Delta}_k, \\ +\infty & \text{otherwise,} \end{cases} \tag{9}$$

*where*

$$\tilde{\Delta}_k(r) \triangleq \left\{ x \,\middle|\, \langle \mathbf{1}, x \rangle \le r, \ 0 \le x_i \le \tfrac{1}{k}, \ i \in [m] \right\}.$$

*Moreover, $\tilde{\phi}_k(a) = \max\{\langle a + c, \lambda \rangle \mid \lambda \in \tilde{\Delta}_k\}$.*

*Proof.* The proof is similar to the proof of Proposition 2; the main step is as follows:

$$\tilde{\phi}_k(a) = \min_{t,\xi,h} \left\{ t + \tfrac{1}{k} \langle \mathbf{1}, \xi \rangle \mid \xi_j \ge h_j - t, \ \xi_j \ge 0, \ h_j \ge a_j + c_j, \ h_j \ge 0 \right\}$$

$$= \max_{\lambda} \left\{ \langle a + c, \lambda \rangle \mid \langle \mathbf{1}, \lambda \rangle \le 1, \ 0 \le \lambda_j \le \tfrac{1}{k} \right\}.$$

$\square$

Note that the upper bounds on $x_i$'s are now fixed to $1/k$, which means the Euclidean projection onto the set $\tilde{\Delta}_k$ is an instance of the continuous quadratic knapsack problem from § 4.1. Unfortunately, the proximal step in the SDCA framework corresponds to a *biased* projection where there is an additional $\ell_2$ regularizer on the sum $\langle \mathbf{1}, x \rangle$ coming from the regularizer in the training objective. To address this issue, we follow the derivation given in the proofs of Lemmas 3 and 5.

The update step for the top-$k$ Usunier loss (5) is equivalent to (with $l = 0$ and $u = 1/k$):

$$\min_{x,s} \left\{ \tfrac{1}{2} \|a - x\|^2 + \tfrac{1}{2}\rho s^2 \mid \langle \mathbf{1}, x \rangle = s, \ s \le r, \ l \le x_i \le u, \ i \in [d] \right\}.$$

Let $t, \lambda \ge 0$, $\mu_i \ge 0$, $\nu_i \ge 0$ be the dual variables, and let $\mathcal{L}$ be the Lagrangian:

$$\mathcal{L} = \tfrac{1}{2} \|a - x\|^2 + \tfrac{1}{2}\rho s^2 + t(\langle \mathbf{1}, x \rangle - s) + \lambda(s - r) - \langle \mu, l\mathbf{1} - x \rangle + \langle \nu, x - u\mathbf{1} \rangle.$$

From the KKT conditions, we have that

$$\partial_x \mathcal{L} = x - a + t\mathbf{1} - \mu + \nu = 0, \qquad \partial_s \mathcal{L} = \rho s - t + \lambda = 0,$$
$$\mu_i(l - x_i) = 0, \qquad \nu_i(x_i - u) = 0, \qquad \lambda(s - r) = 0,$$

which then leads to

$$x = a - t\mathbf{1} + \mu - \nu = \min\{\max\{l, x - t\}, u\}, \qquad \lambda = t - \rho s.$$

Now, we can do case distinction based on the sign of $\lambda$. If $\lambda > 0$, then $\langle \mathbf{1}, x \rangle = s = r$ and $t > \rho r$. In this case $\tfrac{1}{2}\rho s^2 = \tfrac{1}{2}\rho r^2 \equiv \text{const}$, therefore this term can be ignored and we get the knapsack problem from § 4.1. Otherwise, if $s < r$, then $\lambda = 0$ and $t = \rho s$. Using the index sets $U$, $M$ and $L$ as in Lemma 3, we have that

$$t = \rho\left( \sum_L l + \sum_M (a_i - t) + \sum_U u \right) = \rho\left( l\,|L| + u\,|U| - t\,|M| + \sum_M a_i \right).$$

Solving for $t$ with $\rho > 0$, we obtain that

$$t = \left( l\,|L| + u\,|U| + \sum_M a_i \right) \Big/ \left( \frac{1}{\rho} + |M| \right). \tag{10}$$

**Projection.** To compute the (biased) projection, we follow the same steps as in § 4.3. First, we solve the knapsack problem using the algorithm of [14], which also computes the dual variable $t$. If $t > \rho r$, then we are done; otherwise, we sort $a$ and loop over the feasible index sets $U$, $M$, and $L$. We stop once we find a $t$ that satisfies (10) and is compatible with the corresponding index sets.

**Caltech 101 Silhouettes**      **MIT Indoor 67**

| Method | Top-1 | Top-2 | Top-3 | Top-4 | Top-5 | Top-10 | Method | Top-1 | Method | Top-1 | Method | Top-1 |
|---|---|---|---|---|---|---|---|---|---|---|---|---|
| Top-1 [29] | 62.1 | - | 79.6 | - | 83.1 | - | BLH [5] | 48.3 | DGE [7] | 66.87 | RAS [24] | 69.0 |
| Top-2 [29] | 61.4 | - | 79.2 | - | 83.4 | - | SP [28] | 51.4 | ZLX [33] | 68.24 | KL [15] | 70.1 |
| Top-5 [29] | 60.2 | - | 78.7 | - | 83.4 | - | JVJ [13] | 63.10 | GWG [9] | 68.88 | | |

| Method | Top-1 | Top-2 | Top-3 | Top-4 | Top-5 | Top-10 | Top-1 | Top-2 | Top-3 | Top-4 | Top-5 | Top-10 |
|---|---|---|---|---|---|---|---|---|---|---|---|---|
| $\mathrm{SVM}^{\mathrm{OVA}}$ | 61.81 | 73.13 | 76.25 | 77.76 | 78.89 | 83.57 | 71.72 | 81.49 | 84.93 | 86.49 | 87.39 | 90.45 |
| TopPush | 63.11 | 75.16 | 78.46 | 80.19 | 81.97 | 86.95 | 70.52 | 83.13 | 86.94 | 90.00 | 91.64 | 95.90 |
| Prec@1 | 61.29 | 73.26 | 76.12 | 77.76 | 79.11 | 83.27 | 69.03 | 80.67 | 85.00 | 87.16 | 88.21 | 91.87 |
| Prec@5 | 61.73 | 73.99 | 76.90 | 78.50 | 79.63 | 84.22 | 69.18 | 81.42 | 85.45 | 87.61 | 88.43 | 91.87 |
| Prec@10 | 61.90 | 73.95 | 76.68 | 78.46 | 79.67 | 84.14 | 69.18 | 81.42 | 85.45 | 87.61 | 88.43 | 91.87 |
| Recall@1 | 61.55 | 73.13 | 77.03 | 79.41 | 80.97 | 85.18 | 71.57 | 83.06 | 87.69 | 90.45 | 92.24 | 96.19 |
| Recall@2 | 61.25 | 73.00 | 76.33 | 77.94 | 79.15 | 83.49 | 71.42 | 81.49 | 85.60 | 87.24 | 88.36 | 92.16 |
| Recall@3 | 61.51 | 72.95 | 76.55 | 78.72 | 80.49 | 84.74 | 71.42 | 81.57 | 85.67 | 87.39 | 88.43 | 92.24 |
| Recall@4 | 61.55 | 72.95 | 76.68 | 78.80 | 80.58 | 84.70 | 71.42 | 81.57 | 85.67 | 87.24 | 88.28 | 92.01 |
| Recall@5 | 61.60 | 72.87 | 76.51 | 78.76 | 80.54 | 84.74 | 71.49 | 81.49 | 85.45 | 87.24 | 88.21 | 92.01 |
| Recall@10 | 61.51 | 72.95 | 76.46 | 78.72 | 80.54 | 84.92 | 71.42 | 81.49 | 85.52 | 87.24 | 88.28 | 92.16 |
| $\mathrm{W}_{++,\,0/m}$ | 62.33 | 74.95 | 78.59 | 81.45 | 83.66 | 89.08 | 69.33 | 83.06 | 88.66 | 91.72 | 93.43 | 97.54 |
| $\mathrm{W}_{++,\,1/m}$ | 59.69 | 65.97 | 68.92 | 71.61 | 73.82 | 80.88 | 67.39 | 80.15 | 85.22 | 88.88 | 90.90 | 95.90 |
| $\mathrm{W}_{++,\,2/m}$ | 57.39 | 64.33 | 67.88 | 70.13 | 71.95 | 77.59 | 62.61 | 76.57 | 82.39 | 86.19 | 88.36 | 93.81 |
| $\mathrm{W}_{++,\,4/m}$ | 56.78 | 63.94 | 67.36 | 70.05 | 72.08 | 78.76 | 63.13 | 76.87 | 82.24 | 85.67 | 88.43 | 94.63 |
| $\mathrm{W}_{++,\,8/m}$ | 57.17 | 63.50 | 67.01 | 69.79 | 71.87 | 77.85 | 63.73 | 77.24 | 83.36 | 86.87 | 89.10 | 94.63 |
| $\mathrm{W}_{++,\,0/192}$ | 62.29 | 76.25 | 79.71 | 81.40 | 83.09 | 88.17 | 69.78 | 82.99 | 88.36 | 91.49 | 93.51 | 97.31 |
| $\mathrm{W}_{++,\,1/192}$ | 59.56 | 65.97 | 69.44 | 71.65 | 73.91 | 79.45 | 67.24 | 81.34 | 85.60 | 89.03 | 91.19 | 95.75 |
| $\mathrm{W}_{++,\,2/192}$ | 56.78 | 63.29 | 67.10 | 69.87 | 71.69 | 78.37 | 63.28 | 77.61 | 84.03 | 87.99 | 89.93 | 94.85 |
| $\mathrm{W}_{++,\,4/192}$ | 58.13 | 64.37 | 67.62 | 69.92 | 71.56 | 78.15 | 62.54 | 76.79 | 84.10 | 87.61 | 89.18 | 94.03 |
| $\mathrm{W}_{++,\,8/192}$ | 57.04 | 66.28 | 70.18 | 73.39 | 75.34 | 82.79 | 63.06 | 77.84 | 84.55 | 88.06 | 90.37 | 94.70 |
| $\mathrm{W}_{++,\,0/256}$ | 62.68 | 76.33 | 79.41 | 81.71 | 83.18 | 88.95 | 70.07 | 84.10 | 89.48 | 92.46 | 94.48 | **97.91** |
| $\mathrm{W}_{++,\,1/256}$ | 59.25 | 65.63 | 69.22 | 71.09 | 72.95 | 79.71 | 68.13 | 81.49 | 86.64 | 89.63 | 91.42 | 95.45 |
| $\mathrm{W}_{++,\,2/256}$ | 55.09 | 61.81 | 66.02 | 68.88 | 70.61 | 76.59 | 64.63 | 78.43 | 84.18 | 88.13 | 89.93 | 94.55 |
| $\mathrm{W}_{++,\,4/256}$ | 56.52 | 62.29 | 65.76 | 68.01 | 70.13 | 76.59 | 60.90 | 75.97 | 82.84 | 86.79 | 89.63 | 94.63 |
| $\mathrm{W}_{++,\,8/256}$ | 55.79 | 61.60 | 65.58 | 68.23 | 70.39 | 77.55 | 62.39 | 75.15 | 81.42 | 85.82 | 88.88 | 94.03 |
| top-1 $\mathrm{SVM}_\alpha$ | 62.81 | 74.60 | 77.76 | 80.02 | 81.97 | 86.91 | **73.96** | 85.22 | 89.25 | 91.94 | 93.43 | 96.94 |
| top-2 $\mathrm{SVM}_\alpha$ | 63.11 | 76.16 | 79.02 | 81.01 | 82.75 | 87.65 | 73.06 | 85.67 | 90.37 | 92.24 | 94.48 | 97.31 |
| top-3 $\mathrm{SVM}_\alpha$ | **63.37** | 76.72 | 79.67 | 81.49 | 83.57 | 88.25 | 71.57 | **86.27** | **91.12** | 93.21 | 94.70 | 97.24 |
| top-4 $\mathrm{SVM}_\alpha$ | 63.20 | 76.64 | 79.76 | 82.36 | 84.05 | 88.64 | 71.42 | 85.67 | 90.75 | **93.28** | **94.78** | 97.84 |
| top-5 $\mathrm{SVM}_\alpha$ | 63.29 | 76.81 | 80.02 | 82.75 | 84.31 | 88.69 | 70.67 | 85.75 | 90.37 | 93.21 | 94.70 | **97.91** |
| top-10 $\mathrm{SVM}_\alpha$ | 62.98 | **77.33** | 80.49 | 82.66 | 84.57 | 89.55 | 70.00 | 85.45 | 90.00 | 93.13 | 94.63 | 97.76 |
| top-20 $\mathrm{SVM}_\alpha$ | 59.21 | 75.64 | **80.88** | **83.49** | **85.39** | **90.33** | 65.90 | 84.10 | 89.93 | 92.69 | 94.25 | 97.54 |
| top-1 $\mathrm{SVM}_\beta$ | 62.81 | 74.60 | 77.76 | 80.02 | 81.97 | 86.91 | 73.96 | 85.22 | 89.25 | 91.94 | 93.43 | 96.94 |
| top-2 $\mathrm{SVM}_\beta$ | 63.55 | 76.25 | 79.28 | 81.14 | 82.62 | 87.91 | **74.03** | 85.90 | 89.78 | 92.24 | 94.10 | 97.31 |
| top-3 $\mathrm{SVM}_\beta$ | 63.94 | 76.64 | 79.71 | 81.36 | 83.44 | 87.99 | 72.99 | **86.34** | 90.60 | 92.76 | 94.40 | 97.24 |
| top-4 $\mathrm{SVM}_\beta$ | 63.94 | 76.85 | 80.15 | 82.01 | 83.53 | 88.73 | 73.06 | 86.19 | **90.82** | 92.69 | **94.48** | 97.69 |
| top-5 $\mathrm{SVM}_\beta$ | 63.59 | 77.03 | 80.36 | 82.57 | 84.18 | 89.03 | 72.61 | 85.60 | 90.75 | 92.99 | **94.48** | 97.61 |
| top-10 $\mathrm{SVM}_\beta$ | **64.02** | 77.11 | 80.49 | 83.01 | 84.87 | 89.42 | 71.87 | 85.30 | 90.45 | **93.36** | 94.40 | **97.76** |
| top-20 $\mathrm{SVM}_\beta$ | 63.37 | **77.24** | **81.06** | **83.31** | **85.18** | **90.03** | 71.94 | 85.30 | 90.07 | 92.46 | 94.33 | 97.39 |

Table 1: Top-$k$ accuracy (%). **Top section:** State of the art. **Middle section:** Baseline methods. Prec@k and Recall@k are $\mathrm{SVM}^{\mathrm{Perf}}$ [12]; $\mathrm{W}_{++,\,Q/m}$ is $\mathrm{Wsabie}^{++}$ [10] with an embedding dimension $m$ and the queue size $Q$; in the first part, $m = 101$ for Caltech and $m = 67$ for Indoor. **Bottom section:** Top-$k$ SVMs: top-$k$ $\mathrm{SVM}_\alpha$ – with the loss (3); top-$k$ $\mathrm{SVM}_\beta$ – with the loss (5).

## 6 Experimental Results

We have two main goals in the experiments. First, we show that the (biased) projection onto the top-$k$ simplex is scalable and comparable to an efficient algorithm [14] for the simplex projection. Second, we show that the top-$k$ multiclass SVM using both versions of the top-$k$ hinge loss (3) and (5), denoted top-$k$ $\mathrm{SVM}_\alpha$ and top-$k$ $\mathrm{SVM}_\beta$ respectively, leads to improvements in top-$k$ accuracy consistently over all datasets and choices of $k$. In particular, we note improvements compared to the multiclass SVM of Crammer and Singer [6], which corresponds to top-1 $\mathrm{SVM}_\alpha$/top-1 $\mathrm{SVM}_\beta$. We release our implementation of the projection procedures and both SDCA solvers as a C++ library[2] with a Matlab interface.

**SUN 397** (10 splits)

| Top-1 accuracy | XHE [32] | 38.0 | LSH [16] | $49.48 \pm 0.3$ | ZLX [33] | $54.32 \pm 0.1$ |
|---|---|---|---|---|---|---|
| | SPM [26] | $47.2 \pm 0.2$ | GWG [9] | 51.98 | KL [15] | $54.65 \pm 0.2$ |

| Method | Top-1 | Top-2 | Top-3 | Top-4 | Top-5 | Top-10 |
|---|---|---|---|---|---|---|
| $\text{SVM}^{\text{OVA}}$ | $55.23 \pm 0.6$ | $66.23 \pm 0.6$ | $70.81 \pm 0.4$ | $73.30 \pm 0.2$ | $74.93 \pm 0.2$ | $79.00 \pm 0.3$ |
| $\text{TopPush}^{\text{OVA}}$ | $53.53 \pm 0.3$ | $65.39 \pm 0.3$ | $71.46 \pm 0.2$ | $75.25 \pm 0.1$ | $77.95 \pm 0.2$ | $85.15 \pm 0.3$ |
| $\text{Recall@1}^{\text{OVA}}$ | $52.95 \pm 0.2$ | $65.49 \pm 0.2$ | $71.86 \pm 0.2$ | $75.88 \pm 0.2$ | $78.72 \pm 0.2$ | $86.03 \pm 0.2$ |
| $\text{Recall@2}^{\text{OVA}}$ | $52.80 \pm 0.2$ | $64.18 \pm 0.2$ | $68.81 \pm 0.2$ | $71.42 \pm 0.2$ | $73.17 \pm 0.2$ | $77.69 \pm 0.3$ |
| $\text{Recall@3}^{\text{OVA}}$ | $40.50 \pm 0.3$ | $56.01 \pm 0.2$ | $64.96 \pm 0.2$ | $70.95 \pm 0.2$ | $75.26 \pm 0.2$ | $86.32 \pm 0.2$ |
| $\text{Recall@4}^{\text{OVA}}$ | $46.59 \pm 0.4$ | $59.87 \pm 0.6$ | $66.77 \pm 0.5$ | $70.95 \pm 0.4$ | $73.75 \pm 0.3$ | $79.86 \pm 0.2$ |
| $\text{Recall@5}^{\text{OVA}}$ | $50.72 \pm 0.2$ | $64.74 \pm 0.3$ | $70.75 \pm 0.3$ | $74.02 \pm 0.3$ | $76.06 \pm 0.3$ | $80.66 \pm 0.2$ |
| $\text{Recall@10}^{\text{OVA}}$ | $50.92 \pm 0.2$ | $64.94 \pm 0.2$ | $70.95 \pm 0.2$ | $74.14 \pm 0.2$ | $76.21 \pm 0.2$ | $80.68 \pm 0.2$ |
| top-1 $\text{SVM}_\alpha$ | $58.16 \pm 0.2$ | $71.66 \pm 0.2$ | $78.22 \pm 0.1$ | $82.29 \pm 0.2$ | $84.98 \pm 0.2$ | $91.48 \pm 0.2$ |
| top-2 $\text{SVM}_\alpha$ | $58.81 \pm 0.2$ | $72.71 \pm 0.2$ | $79.33 \pm 0.2$ | $83.29 \pm 0.2$ | $85.94 \pm 0.2$ | $92.19 \pm 0.2$ |
| top-3 $\text{SVM}_\alpha$ | $\mathbf{58.97 \pm 0.1}$ | $73.19 \pm 0.2$ | $79.86 \pm 0.2$ | $83.83 \pm 0.2$ | $86.46 \pm 0.2$ | $92.57 \pm 0.2$ |
| top-4 $\text{SVM}_\alpha$ | $58.95 \pm 0.1$ | $73.54 \pm 0.2$ | $80.25 \pm 0.2$ | $84.20 \pm 0.2$ | $86.78 \pm 0.2$ | $92.82 \pm 0.2$ |
| top-5 $\text{SVM}_\alpha$ | $58.92 \pm 0.1$ | $\mathbf{73.66 \pm 0.2}$ | $80.46 \pm 0.2$ | $84.44 \pm 0.3$ | $87.03 \pm 0.2$ | $92.98 \pm 0.2$ |
| top-10 $\text{SVM}_\alpha$ | $58.00 \pm 0.2$ | $73.65 \pm 0.1$ | $\mathbf{80.80 \pm 0.1}$ | $84.81 \pm 0.2$ | $\mathbf{87.45 \pm 0.2}$ | $93.40 \pm 0.2$ |
| top-20 $\text{SVM}_\alpha$ | $55.98 \pm 0.3$ | $72.51 \pm 0.2$ | $80.22 \pm 0.2$ | $84.54 \pm 0.2$ | $87.37 \pm 0.2$ | $\mathbf{93.62 \pm 0.2}$ |
| top-1 $\text{SVM}_\beta$ | $58.16 \pm 0.2$ | $71.66 \pm 0.2$ | $78.22 \pm 0.1$ | $82.29 \pm 0.2$ | $84.98 \pm 0.2$ | $91.48 \pm 0.2$ |
| top-2 $\text{SVM}_\beta$ | $58.80 \pm 0.2$ | $72.65 \pm 0.2$ | $79.26 \pm 0.2$ | $83.21 \pm 0.2$ | $85.85 \pm 0.2$ | $92.14 \pm 0.2$ |
| top-3 $\text{SVM}_\beta$ | $59.14 \pm 0.2$ | $73.21 \pm 0.2$ | $79.81 \pm 0.2$ | $83.77 \pm 0.2$ | $86.36 \pm 0.2$ | $92.51 \pm 0.2$ |
| top-4 $\text{SVM}_\beta$ | $59.24 \pm 0.1$ | $73.58 \pm 0.2$ | $80.18 \pm 0.2$ | $84.15 \pm 0.2$ | $86.71 \pm 0.2$ | $92.73 \pm 0.2$ |
| top-5 $\text{SVM}_\beta$ | $59.28 \pm 0.2$ | $73.78 \pm 0.2$ | $80.45 \pm 0.3$ | $84.36 \pm 0.3$ | $86.96 \pm 0.3$ | $92.93 \pm 0.2$ |
| top-10 $\text{SVM}_\beta$ | $\mathbf{59.32 \pm 0.1}$ | $\mathbf{74.13 \pm 0.2}$ | $80.91 \pm 0.2$ | $84.92 \pm 0.2$ | $87.49 \pm 0.2$ | $93.36 \pm 0.2$ |
| top-20 $\text{SVM}_\beta$ | $58.65 \pm 0.2$ | $73.96 \pm 0.2$ | $\mathbf{80.95 \pm 0.2}$ | $\mathbf{85.05 \pm 0.2}$ | $\mathbf{87.70 \pm 0.2}$ | $\mathbf{93.64 \pm 0.2}$ |

| Method | **Places 205** (val) | | | | | | **ImageNet 2012** (val) | | | | | |
|---|---|---|---|---|---|---|---|---|---|---|---|---|
| | Top-1 | Top-2 | Top-3 | Top-4 | Top-5 | Top-10 | Top-1 | Top-2 | Top-3 | Top-4 | Top-5 | Top-10 |
| ZLX [33] / BVLC [11] | 50.0 | - | - | - | 81.1 | - | **57.4** | - | - | - | 80.4 | - |
| $\text{TopPush}^{\text{OVA}}$ | 38.45 | 47.33 | 53.25 | 57.29 | 60.30 | 69.91 | 55.49 | 68.05 | **73.89** | **77.34** | 79.72 | **85.99** |
| top-1 $\text{SVM}_\alpha$ | 50.63 | 64.47 | 71.44 | 75.50 | 78.54 | 86.17 | **56.61** | 67.31 | 72.43 | 75.45 | 77.67 | 83.71 |
| top-2 $\text{SVM}_\alpha$ | 51.05 | 65.74 | 73.10 | 77.49 | 80.74 | 88.43 | 56.60 | 68.09 | 73.25 | 76.36 | 78.62 | 84.55 |
| top-3 $\text{SVM}_\alpha$ | **51.31** | 66.17 | 73.23 | 77.86 | 81.26 | 89.37 | 56.56 | 68.27 | 73.60 | 76.76 | 79.03 | 84.96 |
| top-4 $\text{SVM}_\alpha$ | 51.24 | **66.30** | 73.48 | 78.08 | 81.40 | 89.74 | 56.52 | 68.36 | 73.80 | 77.06 | 79.30 | 85.25 |
| top-5 $\text{SVM}_\alpha$ | 50.80 | 66.23 | **73.67** | 78.19 | 81.43 | 89.95 | 56.46 | **68.40** | **73.85** | 77.20 | 79.39 | 85.41 |
| top-10 $\text{SVM}_\alpha$ | 50.10 | 65.76 | 73.38 | **78.30** | **81.62** | 90.14 | 55.89 | 68.16 | 73.80 | **77.31** | **79.75** | 85.77 |
| top-20 $\text{SVM}_\alpha$ | 49.25 | 64.85 | 72.62 | 77.67 | 81.14 | 89.99 | 54.94 | 67.53 | 73.50 | 77.08 | 79.59 | **85.88** |
| top-1 $\text{SVM}_\beta$ | 50.63 | 64.45 | 71.45 | 75.50 | 78.54 | 86.17 | 56.61 | 67.31 | 72.43 | 75.45 | 77.67 | 83.71 |
| top-2 $\text{SVM}_\beta$ | 51.03 | 65.58 | 72.73 | 77.40 | 80.55 | 88.40 | 56.91 | 67.98 | 73.19 | 76.23 | 78.50 | 84.43 |
| top-3 $\text{SVM}_\beta$ | 51.27 | 65.98 | 73.37 | 77.91 | 81.25 | 89.30 | 57.00 | 68.27 | 73.51 | 76.68 | 78.89 | 84.84 |
| top-4 $\text{SVM}_\beta$ | **51.38** | 66.20 | 73.56 | 78.04 | 81.40 | 89.78 | 56.99 | 68.39 | 73.62 | 76.86 | 79.15 | 85.09 |
| top-5 $\text{SVM}_\beta$ | 51.25 | **66.25** | **73.66** | 78.26 | 81.42 | 89.91 | **57.09** | **68.45** | 73.68 | 76.95 | 79.27 | 85.24 |
| top-10 $\text{SVM}_\beta$ | 50.94 | 66.13 | 73.52 | **78.36** | **81.69** | **90.19** | 56.90 | 68.42 | **73.95** | 77.31 | 79.53 | 85.62 |
| top-20 $\text{SVM}_\beta$ | 50.50 | 65.79 | 73.38 | 78.17 | 81.60 | 90.12 | 56.48 | 68.29 | 73.83 | **77.32** | **79.60** | 85.81 |

Table 2: Top-$k$ accuracy (%). **Top section:** State of the art. **Middle section:** Baseline methods. **Bottom section:** Top-$k$ SVMs: top-$k$ $\text{SVM}_\alpha$ – with the loss (3); top-$k$ $\text{SVM}_\beta$ – with the loss (5). Results for Places 205 and ImageNet 2012 are computed on the validation set.

## 6.1 Scaling of the Projection onto the Top-$k$ Simplex

We follow the experimental setup of [18]. We sample 1000 points from the normal distribution $\mathcal{N}(0,1)$ and solve the projection problems using the algorithm of [14] (denoted as Knapsack) and using our proposed method of projecting onto the set $\Delta_k$ for different values of $k = 1, 5, 10$. We report the total CPU time taken on a single Intel(R) Xeon(R) 2.20GHz processor. As one can see, the scaling is linear in the problem dimension and the run times are essentially the same.

Figure 3: Scaling of the projection onto the top-$k$ simplex compared to the knapsack problem.

### 6.2 Image Classification Experiments

We evaluate our method on five image classification datasets of different scale and complexity: Caltech 101 Silhouettes [29] ($m = 101$, $n = 4100$), MIT Indoor 67 [23] ($m = 67$, $n = 5354$), SUN 397 [32] ($m = 397$, $n = 19850$), Places 205 [33] ($m = 205$, $n = 2448873$), and ImageNet 2012 [25] ($m = 1000$, $n = 1281167$). For Caltech, $d = 784$, and for the others $d = 4096$. The results on the two large scale datasets are in the supplement.

We cross-validate hyper-parameters in the range $10^{-5}$ to $10^3$, extending it when the optimal value is at the boundary. We use LibLinear [8] for $\text{SVM}^{\text{OVA}}$, $\text{SVM}^{\text{Perf}}$ [12] with the corresponding loss function for $\text{Recall@k}$, and the code provided by [17] for TopPush. When a ranking method like $\text{Recall@k}$ and TopPush does not scale to a particular dataset using the reduction of the multiclass to a binary problem discussed in § 2.3, we use the one-vs-all version of the corresponding method. We implemented Wsabie$^{++}$ (denoted $\text{W}_{++}$, $\text{Q/m}$) based on the pseudo-code from Table 3 in [10]. Among the baseline methods that we tried, only $\text{TopPush}^{\text{OVA}}$ scaled to the Places and the ImageNet datasets both time and memory-wise[3].

On Caltech 101, we use features provided by [29]. For the other datasets, we extract CNN features of a pre-trained CNN (fc7 layer after ReLU). For the scene recognition datasets, we use the Places 205 CNN [33] and for ILSVRC 2012 we use the Caffe reference model [11].

Experimental results are given in Tables 1, 2. First, we note that our method is scalable to large datasets with millions of training examples, such as Places 205 and ILSVRC 2012 (results in the supplement). Second, we observe that optimizing the top-$k$ hinge loss (both versions) yields consistently better top-$k$ performance. This might come at the cost of a decreased top-1 accuracy (e.g. on MIT Indoor 67), but, interestingly, may also result in a noticeable increase in the top-1 accuracy on larger datasets like Caltech 101 Silhouettes and SUN 397. This resonates with our argumentation that optimizing for top-$k$ is often more appropriate for datasets with a large number of classes.

Overall, we get systematic increase in top-$k$ accuracy over all datasets that we examined. For example, we get the following improvements in top-5 accuracy with our top-10 $\text{SVM}_\alpha$ compared to top-1 $\text{SVM}_\alpha$: $+2.6\%$ on Caltech 101, $+1.2\%$ on MIT Indoor 67, and $+2.5\%$ on SUN 397.

## 7  Conclusion

We demonstrated scalability and effectiveness of the proposed top-$k$ multiclass SVM on five image recognition datasets leading to consistent improvements in top-$k$ performance. In the future, one could study if the top-$k$ hinge loss (3) can be generalized to the family of ranking losses [30]. Similar to the top-$k$ loss, this could lead to tighter convex upper bounds on the corresponding discrete losses.

## Footnotes

[1] A convex function $f : X \to \mathbb{R} \cup \{\pm\infty\}$ has an *effective domain* $\operatorname{dom} f = \{x \in X \mid f(x) < +\infty\}$.

[2] https://github.com/mlapin/libsdca