[Reviews · NeurIPS 2015]

Submitted by Assigned_Reviewer_1

The paper proposes a top-k hinge loss for top-k classification. It generalizes the multiclass SVM and uses the sum of k largest components as a convex approximation to the desired top-k loss. An efficient optimization algorithm is proposed and relationships with previous work are discussed. Paper is clearly written. Experiment results show the effectiveness of the proposed method compared to baselines.

1. The authors did not report comparisons with [25, 8] due to availability of the implementations. However, I am really interested in seeing the comparisons.

2. Can the authors make the code public upon acceptance?

Summary: A neat formulation of top-k multiclass SVM that works well.

Submitted by Assigned_Reviewer_2

In many multiclass classification problems with a large number of classes, semantic similarity between classes can typically be observed. This paper proposed an algorithm called top-k SVM to explicitly deal with the learning scenario where you only get a cost when the ground truth is outside the top k proposed classes (e.g. imagenet challenge). After extending the loss function to top-k loss, an efficient optimization procedure is formulated. Experiments on large scale datasets showed the scalability of the algorithm and clear improvement could be observed over compared algorithms.

The paper is clearly motivated and easy to follow. And the problem (similar classes in multiclass learning) they work on is an important one that happens a lot in many practical applications.
Summary: This paper formulated the top-k SVM algorithm to handle the case in multiclass classification when some classes might be very similar to each other. The algorithm scales to very large dataset and showed improvements over compared methods.

Submitted by Assigned_Reviewer_3

The material in this paper is non-trivial and the paper is well-written. The experimental results are also good.

The only suggestion I have is to motivate the choice of the loss function further.

Also, I spotted a typo in page 3 - "globally optimal leads to" (remove optimal).
Summary: This is a nice paper that describes an interesting problem. The idea to to learn with respect to the top-k loss, where k predictions are made and the loss is zero if the true class is included in the k predictions.

Author Feedback
Author rebuttal: We thank all the reviewers for their helpful and positive feedback. We are happy that all the reviewers acknowledge the contributions of this work, in particular, a novel loss function that addresses a relevant practical problem, an efficient optimization scheme that scales to large datasets, and an experimental evaluation that shows encouraging results.

Reviewer 2:
- motivate the choice of the loss function further.

We provide two possible directions to extend the motivation.

First, the top-k error is used as the performance measure in certain benchmarks, e.g. in the ImageNet challenge in computer vision. Therefore, it makes sense to directly optimize it.

Second, if one is using linear classifiers and classes overlap, then optimizing the top-1 loss or using a reduction scheme such as one-vs-all leads to suboptimal results as we demonstrate in our experiments. Our explanation is that it is difficult to discriminate the overlapping classes with a linear classifier. This problem is reduced when optimizing a top-k loss (also seen from our results) and we often even improve the top-1 performance (while consistently improving the top-k accuracy).

- a typo in page 3
Will be fixed.

Reviewer 3:
- relevant recent work (Kar et al., ICML 2015).

We thank the reviewer for the reference to this very recent work and are happy to cite it in the final paper. Please note, however, that Kar et al. consider prec@k, whereas the multiclass top-k error that we consider corresponds to recall@k via the reduction scheme that we discuss in lines 177-191.

Reviewer 5:
- comparisons with Usunier et al., Gupta et al.

Please note that there is no code available for both papers.

In the meantime, we have our own Matlab implementation of Wsabie++ based on the pseudo-code from Table 3 in Gupta et al. We have obtained results on two (smaller) datasets - Caltech101 and MIT Indoor 67. Currently, the experiments do not scale due to Matlab and since Wsabie++ requires to optimize 4 hyperparameters (embedding dimension m, stepsize, margin, depth of the violator chain Q). We obtained the best results with the largest m = 256 (similar to Gupta et al.) and the smallest Q = 0, which could mean that these datasets are still too small for the violator chain to become useful.

On Caltech101, Wsabie++ with Q = 0 obtains top-1 accuracy comparable to the SVM Top-1, and then improves by 1-2% in top-2..10 metrics (which is below our proposed SVM Top-k result). On MIT Indoor 67, Wsabie++ underperforms in top-1..2 metrics, then it is between SVM Top-1 and SVM Top-5 in top-3..4 metrics, and finally it is comparable to SVM Top-3..5 in top-5..10 metrics. We will try to obtain also the results on the larger datasets and include them in the final version.

Regarding Usunier et al, we have derived in the meantime the corresponding conjugate function of the loss to integrate it into our SDCA framework. We are working on the implementation so that it should be possible to report results in the final version.

- publish code upon acceptance?

The source code of our method will be published on GitHub along with the supplementary projection procedures.